# *Toxoplasma gondii* peptide ligands open the gate of the HLA class I binding groove

Curtis McMurtrey[1,2], Thomas Trolle[3,4], Tiffany Sansom[5], Soumya G Remesh[4], Thomas Kaever[4], Wilfried Bardet[1], Kenneth Jackson[1], Rima McLeod[6], Alessandro Sette[4], Morten Nielsen[3,7], Dirk M Zajonc[4], Ira J Blader[5], Bjoern Peters[4], William Hildebrand[1,2]*

[1]Department of Microbiology and Immunology, University of Oklahoma Health Sciences Center, Oklahoma City, United States; [2]Pure MHC LLC, Austin, United States; [3]Center for Biological Sequence Analysis, Technical University of Denmark, Kongens Lyngby, Denmark; [4]La Jolla Institute for Allergy and Immunology, La Jolla, United States; [5]Department of Microbiology and Immunology, University at Buffalo School of Medicine, Buffalo, United States; [6]University of Chicago, Chicago, United States; [7]Instituto de Investigaciones Biotecnológicas, Universidad Nacional de San Martín, Buenos Aires, Argentina

**Abstract** HLA class I presentation of pathogen-derived peptide ligands is essential for CD8+ T-cell recognition of *Toxoplasma gondii* infected cells. Currently, little data exist pertaining to peptides that are presented after *T. gondii* infection. Herein we purify HLA-A*02:01 complexes from *T. gondii* infected cells and characterize the peptide ligands using LCMS. We identify 195 *T. gondii* encoded ligands originating from both secreted and cytoplasmic proteins. Surprisingly, *T. gondii* ligands are significantly longer than uninfected host ligands, and these longer pathogen-derived peptides maintain a canonical N-terminal binding core yet exhibit a C-terminal extension of 1–30 amino acids. Structural analysis demonstrates that binding of extended peptides opens the HLA class I F' pocket, allowing the C-terminal extension to protrude through one end of the binding groove. In summary, we demonstrate that unrealized structural flexibility makes MHC class I receptive to parasite-derived ligands that exhibit unique C-terminal peptide extensions.

*For correspondence: william-hildebrand@ouhsc.edu

**Competing interests:** The authors declare that no competing interests exist.

## Introduction

CD8 T-cells mediate immunity to *Toxoplasma gondii* infection (*Khan et al., 1988*; *Suzuki and Remington, 1988*) through recognition of peptide antigens presented by the MHC class I (MHC I) molecules of infected cells (*Brown and McLeod, 1990*; *Deckert-Schlüter et al., 1994*). The majority of peptide ligands identified to date are derived from parasite surface proteins, proteins localized to dense granules, or the rhoptry proteins which are specialized secretory granules whose contents are released either into the host cell cytoplasm or the parasitophorous vacuole (*Blanchard et al., 2008*; *Cardona et al., 2015*; *Cong et al., 2011*). These secreted proteins are thought to be optimal candidates for MHC I presentation because they have the best access to conventional antigen processing and presentation machinery in the host cell. However, this is a large pathogen, and the full array of parasite proteins that might be sampled and presented remains unknown.

Recent advances in immunology and proteomics highlight that non-canonical ligands are presented to T cells by MHC I molecules. While a majority of peptides are 8–11 amino acids in length, MHC I molecules present a considerable number of peptides >11 amino acids (*Hassan et al., 2015*;

**eLife digest** *Toxoplasma gondii* is a parasite that can infect most warm-blooded animals and cause a disease called toxoplasmosis. In humans, toxoplasmosis generally does not cause any noticeable symptoms, but it can cause serious problems in pregnant women and individuals with weakened immune systems.

*T. gondii* is one of many parasites that hide within human cells in an attempt to avoid detection by the immune system. However, proteins called Human Leukocyte Antigens, or HLAs, can reveal hidden parasites by carrying small sections of them from the inside the infected cell to the cell's surface. The immune system can then recognize the fragments as foreign and attack the parasite.

HLAs typically pick up parasite fragments of a certain length, which enables the immune system to recognize that what is being displayed is a piece of parasite. By purifying HLAs from cells that have been infected by *T. gondii*, McMurtrey et al. have now learned more about which fragments of the parasite are displayed to the immune system. This analysis revealed that the parasite somehow manipulates the HLAs to carry parasite fragments that are considerably longer than can be explained with our current knowledge of how HLAs work. By using a technique called X-ray crystallography, McMurtrey et al. also show that the structure of the HLA assumes a previously unseen configuration when interacting with fragments of *T. gondii*.

In the future, it will be important to understand how infected cells give rise to unusual structural configurations of HLAs and to unravel how these structures affect the immune system's ability to fight infections.

*Schittenhelm et al., 2014*) that elicit T-cell responses (*Hassan et al., 2015*; *Burrows et al., 2006*). Structural characterizations suggest that these long ligands interact with the MHC I molecule much like canonical peptides: The MHC I alpha chain forms a 10 x 25 angstrom groove in which peptide ligands are anchored by their second (P2) and C-terminal (PΩ) residues. In this mode of binding, the middle portion of any oversized peptides can bulge out of the MHC I groove and interact with the receptors of T lymphocytes (*Tynan et al., 2005*). Crystallographic studies have confirmed this bulging model, although there exists a structural example of a 10mer interacting with MHC I molecule HLA-A2 via P2 and P9 with an amino acid extension at P10 (*Collins et al., 1994*). Thus, both peptide extension and peptide bulging have been observed for MHC I ligands, and, as longer ligands become increasingly evident, the interaction of these ligands with MHC I will need to be clarified.

The goal of this study was to have the MHC I of infected cells inform the number, breadth, and nature of *T. gondii* peptide ligands. HLA-A*02:01 was purified from cells infected with *T. gondii* and peptide ligands eluted from the HLA class I (human MHC I) complex were analyzed by two-dimensional LCMS. The resulting data demonstrate that nearly 200 ligands originating from close to 100 different *T. gondii* proteins are sampled for MHC I presentation. As envisioned, a number of ligands originating from dense granule proteins was observed (*Blanchard et al., 2008*; *Cong et al., 2011*), yet MHC I ligands were also derived from a large number parasite cytoplasmic proteins. Surprisingly, *T. gondii* ligands were significantly longer than existing structural models can accommodate, and a series of peptide analogs demonstrated that these longer peptides are not anchored to MHC I via their C-termini. Crystallographic studies reveal an unreported structural re-arrangement of residues in the MHC I binding groove that accommodate C-terminal peptide extensions, and this structural flexibility is discussed in the context of infection by intracellular pathogens.

## Results

### Identification of *Toxoplasma gondii* HLA-A*02:01 ligands

The first objective of this study was to identify pathogen-encoded ligands made available by MHC I. To accomplish this objective, HLA-A*02:01 was purified from *T. gondii* infected THP-1 monocytes as described. (*McMurtrey et al., 2008*; *Wahl et al., 2009*). To ensure THP-1 cells were infected, the number of infected cells and free parasites were periodically assessed. Over the course of a 1-week infection, the number of infected cells increased from 12.1% day 1 post infection to 71.5% on day 7

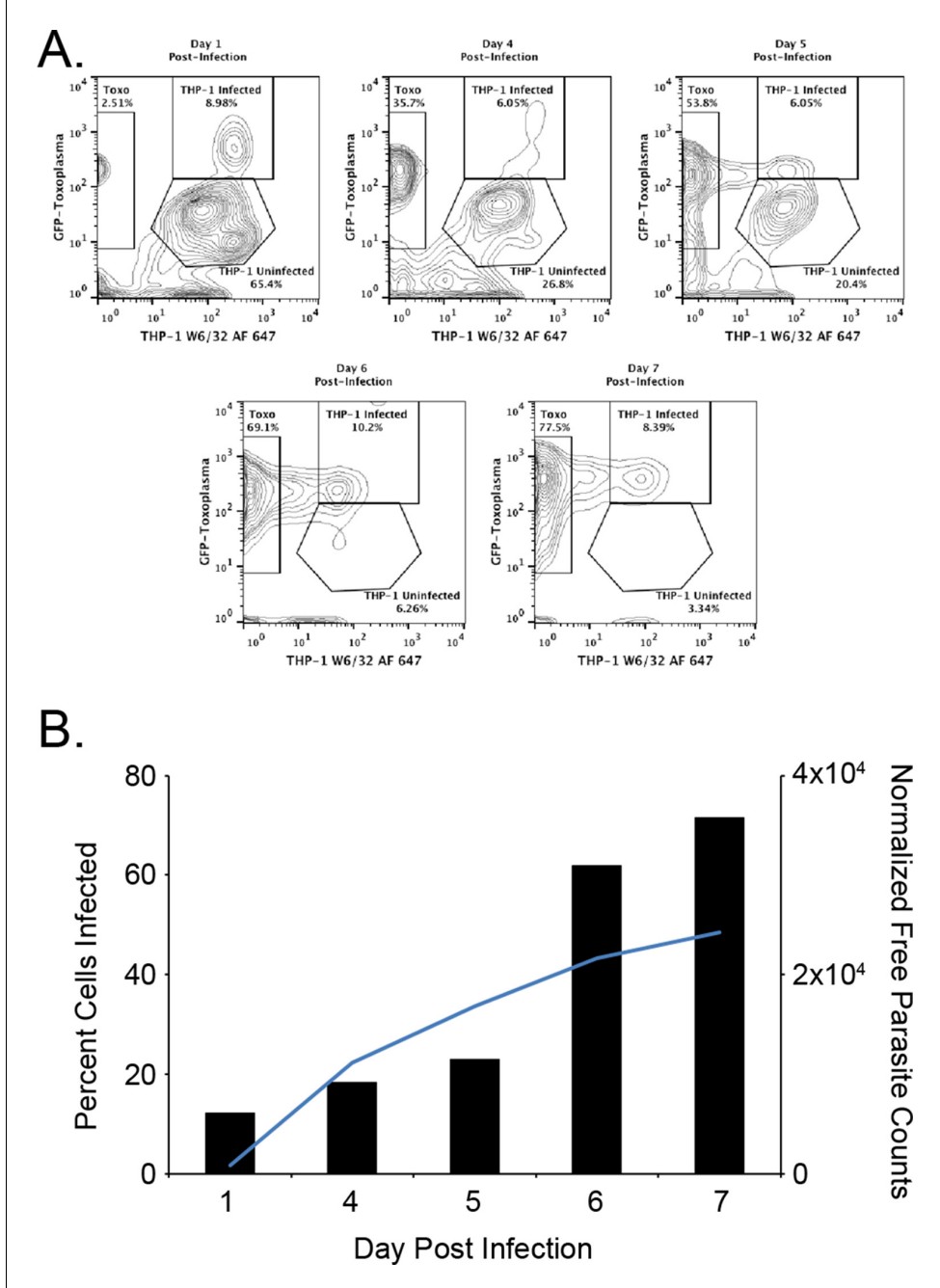

**Figure 1.** Kinetics of the *T. gondii* infection in the bioreactor production. (A) Raw flow cytometry data and gates of the samples taken from the bioreactor on each indicated day post infection. (B) Histogram of the percent of infected cells (black bars) as well as the normalized free parasite counts (blue line). Raw parasite counts were normalized to the total counts of each respective experiment.

post infection (*Figure 1*). A steady increase in the number of free parasites in the culture media from day 1 to 7 was indicative of a productive infection. The production of 12 mg HLA-A*02:01 from infected cells was sufficient for a comprehensive analysis of *T. gondii* peptide ligands.

Peptide ligands eluted from HLA complexes were separated by offline HPLC fractionation and subjected to nanoLCMS. A total of 284 peptide sequences precisely matched the reported sequence of *T. gondii*. Strikingly, 89 of these peptides were either an exact match or contained an isobaric Ile/ Leu ambiguity match to sequences of the *H. sapiens* host species (*Supplementary file 1*), leaving

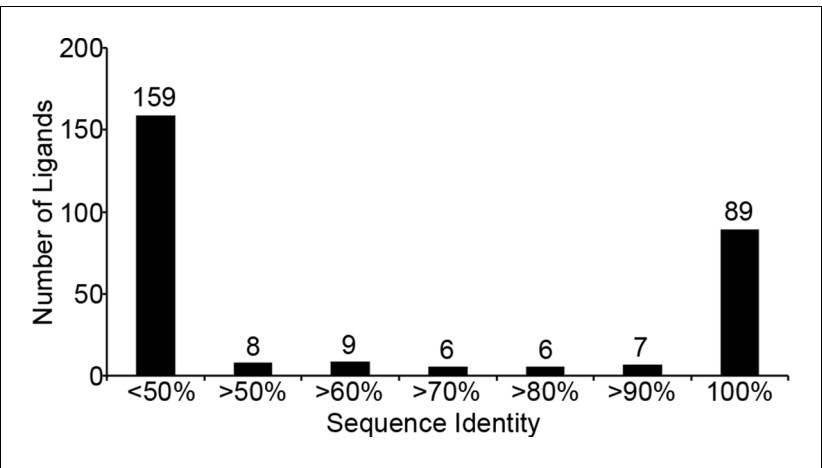

**Figure 2.** Sequence identity of identified *T. gondii* ligands to *H. sapiens*. *T. gondii* derived sequences were BLAST searched against the NCBInr *H. sapiens* proteome. Sequence identity was recorded and ligands with <50% sequence identity were considered not significant and were binned together.

195 ligands definitively derived from *T. gondii* (*Supplementary file 2*). Of these confirmed *T. gondii* derived ligands, most (159) show little (<50%) similarity to *H. sapiens* while 36 peptides have >50% similarity. In summary, 31% of the *T. gondii* derived ligands are identical to *H. sapiens* sequences, 13% show host similarity, and 56% of the *T. gondii* ligands have no similarity to host proteins (*Figure 2*). MHC I makes hundreds of parasite-encoded ligands available to the immune system.

## Source proteins of *T. gondii* ligands

Previous studies have focused upon dense granule proteins (GRA) that are secreted by *T. gondii* into the cytosol of the host cell, and a handful of GRA ligands have been reported (*Blanchard et al., 2008*; *Cardona et al., 2015*; *Cong et al., 2011*; *2012*). In our dataset, the 195 *T. gondii* ligands identified originate from 95 different proteins of which 87 are non-GRA proteins, demonstrating that a considerable number of proteins are accessible to MHC I presentation. For 55.8% of the *T. gondii* source proteins a single peptide was presented (*Figure 3A*), while elongation factor 1 alpha provided 12 peptide ligands. Peptide ligand enrichment from particular proteins was not due to protein length as median protein lengths were not statistically different (Kruskal-Wallis test, p = 0.247) regardless of the number of ligands embedded within a protein (*Figure 3—figure supplement 1*). Amongst the proteins sampled more than once, a hierarchy emerged whereby several hypothetical proteins were most frequently sampled followed by the dense granule proteins, ribosomal proteins, EF1α, tRNA synthetases, and HSP70, respectively (*Figure 3B*, *Supplementary file 2*). All together, multiply sampled *T. gondii* source proteins provided 44.2% of the pathogen-derived ligands. Dense granule proteins have been reported as a source of peptides (*Blanchard et al., 2008*; *Cardona et al., 2015*; *Cong et al., 2010*; *2011*; *2012*; *El Bissati et al., 2014*), and here 30 GRA ligands (15%) were observed with GRA12 providing the most (*Hassan et al., 2015*) peptide ligands (*Figure 3C*). The secreted GRA proteins represent a minority protein source in the rich ligand landscape of this pathogen.

In the search for trends and biases in *T. gondii* peptide ligands, we tested for significant enrichments in cellular compartments of sampled source proteins. There was a significant enrichment in proteins localized to the apical part of the *T. gondii* cell (GO:0045177, p = 1.50 x 10$^{-8}$) and to the parasitophorous vacuole (GO:0020003, p = 2.79 x 10$^{-6}$). Unexpectedly, there was a significant enrichment in proteins originating from the parasite cytoplasm (GO:0005737, p = 9.31 x 10$^{-4}$) with 21 of 95 proteins annotated as cytoplasmic. This enrichment in peptides derived from *T. gondii* cytoplasm proteins shows that *T. gondii* does not sequester cytoplasmic proteins from host MHC I antigen processing and presentation.

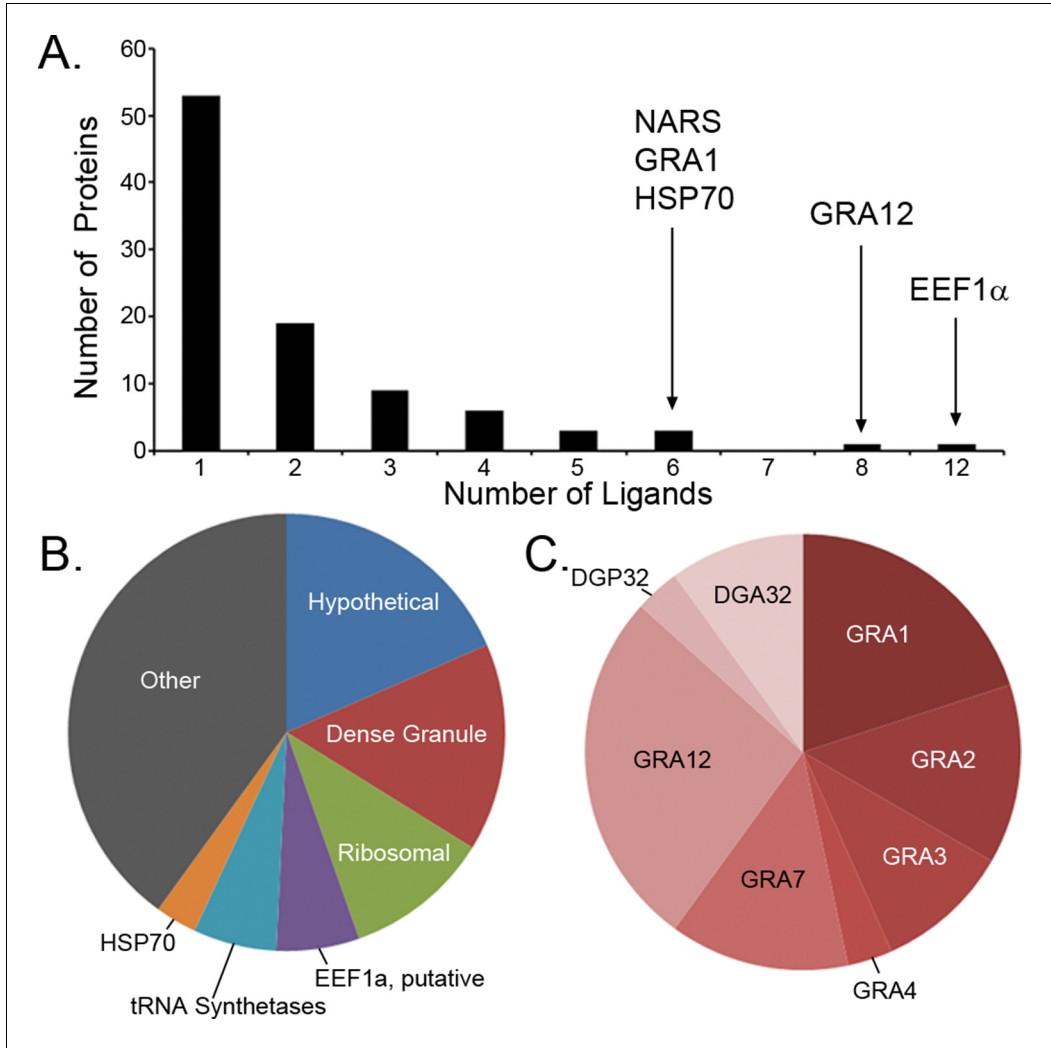

**Figure 3.** Ligand sampling of source proteins. (**A**) The number of distinct ligands from a given source protein was counted binned by number of ligands. Gene symbols of the most sampled proteins are shown above the respective bin. (**B**) Distribution of ligands by source protein group or individual source protein. (**C**) Distribution of ligands by source dense granule protein.
The following source data and figure supplement are available for figure 3:

**Source data 1.** PEAKS export file containing HLA-A*02:01 peptide *H. sapiens* derived ligands from uninfected THP-1 cells.
**Figure supplement 1.** Number of ligands do not correspond to source protein length.

## *T. gondii* ligands are enriched from the C-terminal end of the source protein

A recent study showed that the C-terminal location of an epitope within the source protein was important for *T. gondii* ligand presentation and immunodominance (*Feliu et al., 2013*). Given this observation, we assessed the location of ligands within their source proteins to determine if a C-terminal bias was maintained. Normalized ligand position was calculated as described by Kim *et al.* (*Kim et al., 2013*) and were binned into 5 positions with the most N-terminal bin being 0–0.2 and the most C-terminal bin being 0.8–1.0 (*Figure 4*). When the *T. gondii* ligands were compared to the baseline uninfected ligand distribution there was a significant reduction in N-terminal peptides (bins 0–0.2, p = 2.36 x 10$^{-4}$ and 0.2–0.4, p = 2.23 x 10$^{-4}$, comparison of proportions) along with significant

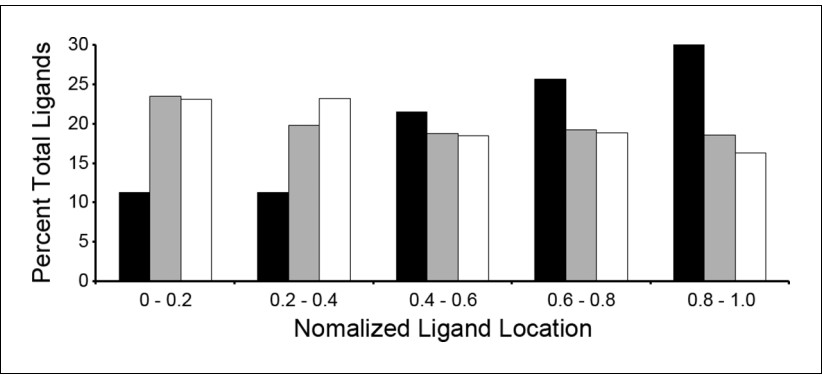

**Figure 4.** Location of ligands within respective source proteins. Normalized ligand location within the respective source protein from the unambiguous *T. gondii* ligands (black), *H. sapiens* ligands from infected THP-1 cells (white) and, *H. sapiens* ligands from uninfected THP-1 cells (grey).

enrichment in ligands from the C-terminal end of the protein (bins 0.6–0.8, p = 1.46 x 10$^{-2}$ and 0.8–1.0, p = 5.26 x 10$^{-7}$). There was no significant change (p = 0.238) to the central ligands (bin 0.4–0.6). Next, host-derived ligands from the infected and uninfected cells were compared to see if the C-terminal bias of pathogen encoded peptides extended to the infected host. There was significant reduction in the 0.2–0.4 bin (p = 0.007) with a significant increase (p = 0.0095) in the very C-terminal bin (p = 0.0095). In summary, *T. gondii* ligands are significantly enriched from the C-terminal end of their source proteins and host-derived peptide ligands shift towards the C-termini of their respective source proteins following infection.

## *T. gondii* ligands are significantly longer than host ligands

Our in depth proteomics approach identified thousands of HLA-A*02:01 peptide ligands of a canonical length of 8–11 as well as ligands of non-canonical length. Noteworthy is that a number of the *T. gondii* ligands were much longer than expected for the HLA-A*02:01 molecule (*Figure 5*). When compared with the host-derived ligands, the *T. gondii* ligands were significantly longer (t-test,

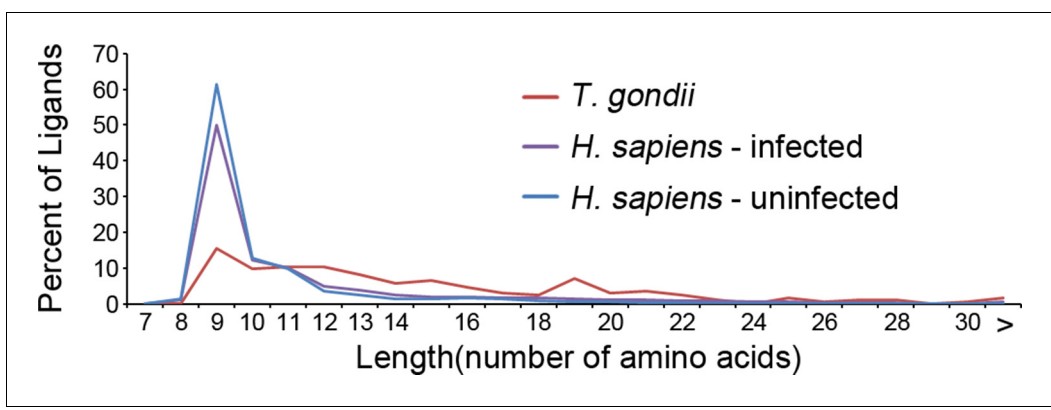

**Figure 5.** Length distribution of identified ligands. Length distributions of unambiguous *T. gondii* ligands (red), *H. sapiens* ligands from infected THP-1 cells (purple) and, *H. sapiens* ligands from uninfected THP-1 cells (blue).
The following source data is available for figure 5:

**Source data 1.** PEAKS export file containing HLA-A*02:01 peptide *T. gondii* derived ligands from *T. gondii* infected THP-1 cells.
**Source data 2.** PEAKS export file containing HLA-A*02:01 peptide *H. sapiens* derived ligands from *T. gondii* infected THP-1 cells.

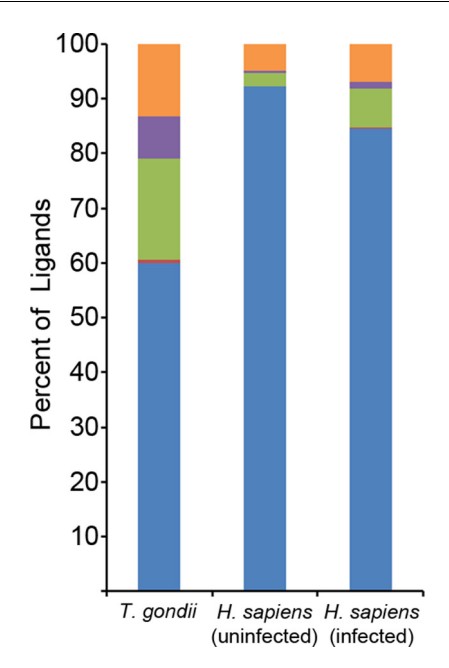

**Figure 6.** Binding prediction analysis of eluted ligands. Percentage of total ligands in indicated dataset that are predicted to be canonical binders (blue), contain a C-terminal binding core (red), contain an N-terminal binding core (green), contain a central binding core (purple), or not predicted to bind (orange).

p<0.001) with an average length of 14.6 amino acids compared with 11.4 amino acids in the host ligands from the infected cells and with 9.8 for uninfected host ligands. This increase in host ligand length following infection was statistically significant (t-test, p<0.001). Thus, infection increases the length of *T. gondii* and host-derived ligands.

## *T. gondii* ligands contain a C-terminal extension from an N-terminal binding core

To investigate how long peptides observed following infection might bind to MHC I, in silico algorithms were used to predict HLA-A*02:01 ligand binding affinity. NetMHCpan 2.8 predicted that 117 of 195 ligands (60%) bind at a percentile rank score <=10%, leaving 78 predicted non-binders (*Figure 6*). As these long peptides were purified from the HLA-A*02:01 of infected cells, we hypothesized that the predicted non-binders interact with HLA-A*02:01 in a non-canonical fashion that escapes algorithms trained on canonical binding data. A subsequent search revealed nested core sequences of 8–11 amino acids within the longer peptides that were not predicted to bind, and these cores were predicted to bind to HLA-A*02:01. Strikingly, a considerable number (52/78) of the predicted non-binders contained nested core sequences of 8–11 amino acids that were predicted to bind with high affinity (percentile rank score <=2%). This was significantly more (p = 1.2 x $10^{-10}$, comparison of proportions) than if the sequences were randomly scrambled (13/78). Hence, a binding core appropriate in length and sequence for interaction with HLA-A*02:01 was embedded within long *T. gondii* peptides.

We next assessed the positioning of binding cores within longer ligands. Within the *T. gondii* ligands, most (36/52) binding cores were at the very N-terminus with the peptide extending from the core's C-terminus (*Supplementary file 3*). This occurred significantly more than randomly scrambled versions of the peptides (1/13) (p = 3.1 x $10^{-5}$, comparison of proportions). There was no significant increase in proportions of ligands with an N-terminal extension (1/52) or an extension on both sides (15/52) compared to scrambled ligand sequences (0/13 and 12/13 respectively). Noteworthy was that 10/15 of the peptides with predicted extensions on both sides of a central core had an alternate binding core at the N-terminal side with a predicted affinity slightly weaker than the central binding core (*Supplementary file 3*). So, most peptides could bind either at the N-terminal end or in the center, and many of the ligands with central cores possessed an alternate N-terminal core. Among the N-terminal binders, the average C-terminal extension was 8.7 amino acids and the extension varied considerably from 1 to 30 amino acids (*Supplementary file 3*). In summary, most of the long *T. gondii* predicted non-binders had an N-terminal binding core with a considerable C-terminal extension.

The observation of C-terminally extended peptides among the *T. gondii* derived ligands prompted a similar analysis of the host ligand repertoire from infected cells. Among ligands from uninfected cells, 2.3% had a C-terminal extension - the baseline of extended peptides in uninfected THP-1 cells. After *T. gondii* infection, the percentage of host ligands with a C-terminal extension significantly increased to 7.2% (p<0.0001, comparison of proportions) (*Figure 6*), a percentage that is less than half the *T. gondii* derived ligands with extensions (18.5%). Thus, *T. gondii* infection results

in C-terminally extended host ligands being nearly 3 times more frequent and *T. gondii* C-terminally extended ligands being eight times more common.

## Predicted extended *T.gondii* ligands bind HLA

The observation of nested binding cores at the N-termini of extended ligands suggests a novel interaction of *T. gondii* peptides and MHC I. To confirm that the N-terminal portion of the peptides were binding to HLA-A*02:01, we synthesized full-length peptides and their corresponding binding cores and determined their affinities for HLA-A*02:01 in a competitive binding assay. Twelve ligands having binding cores with the highest predicted binding affinities (*Supplementary file 3*, bold) were selected for testing in this manner. Of these 12 peptides, eight binding cores bound with high affinity (<500 nM), three with moderate affinity (<1000 nM), and one had no affinity – the N-terminal cores overwhelmingly bound to HLA-A*02:01. Two of the twelve full-length ligands (YLSPIASP-LLDGKSLR-RPL7A[15-30]) and FVLELEPEWTVK-UFP[16-27]) also bound with high affinity (*Figure 7A*) even though predictions ranked them as non-binders as their C-termini were incompatible with the HLA-A*02:01 binding motif. The remaining 10 long peptides did not bind in this in vitro assay, perhaps because they are incapable of binding in the absence of molecular chaperones or the distinct conditions of the infected cell. These data confirm that the N-termini of extended peptides have a strong affinity for MHC I.

Next, the two extended peptides that bound in the in vitro binding assay were tested to see if the binding core of the full-length ligand was essential for binding. In order to assess the contribution of the putative core for peptide binding, HLA-A*02:01 binding assays were completed using the full length peptide, the binding core only, and a series of full length peptide variants containing non-permissive amino acid substitutions at the putative C-terminal anchor of the binding core. For F-VLELEPEWTVK, the affinity of all 'mutant' peptides was significantly worse (t-test, p<0.05) than either the binding core or the native full-length peptide (*Figure 7B*). With YLSPIASPLLDGKSLR, all but one (L10E) of the mutants had a lower binding affinity in comparison to the wild type sequence (t-test, p<0.05) (*Figure 7B*). As YLSPIASPLLDGKSLR harbors a Leu at P9 and at P10, it could be that either Leu can serve as an F' pocket anchor for HLA-A*02:01 and when the P10 was changed to an acidic residue like L10E, the P9 Leu was used as the F-pocket anchor. In summary, the binding core's F-pocket residue is critical for long-peptide binding to HLA-A*02:01 such that C-terminal amino acid extensions of these longer ligands somehow protrude out of the groove in the vicinity of the F-pocket.

## Thermostability of extended ligand complexes

Extended ligands YLSPIASPLLDGKSLR-RPL7A[15-30] and FVLELEPEWTVK-UFP[16-27] bound at high affinity to HLA-A*02:01, and we next assessed their relative stability when in complex with MHC I. To determine the relative stability of these extended peptide/MHC I complexes, we completed a thermal denaturation assay on the extended peptides and their respective binding cores in complex with HLA-A*02:01 (*Figure 8*). The melting temperature (Tm) for the two extended ligands exceeded 62°C (YLSPIASPLLDGKSLR, Tm = 66° C, FVLELEPEWTVK Tm = 63°C), temperatures that are consistent with ligands of conventional length (*Hassan et al., 2015*). The Tm of these extended ligands are also within the range of 15 mers that are reported to 'bulge' in the central portion of HLA-A*02:01 binding groove (*Hassan et al., 2015*). It was somewhat surprising that these extended ligands had Tm representative of canonical ligands, and more remarkable was that the thermostability of the binding cores was at least 10°C higher (YLSPIASPLLDGKSLR, ΔTm = 10°C, FVLELEPEWTVK ΔTm = 12°C) than their extended counterparts (*Figure 8*). Thus, extended ligands have a denaturation temperature consistent with those of conventional ligands, possibly due to the highly thermostable nature of their binding cores.

## The C-terminus of extended ligand FVLELEPEWTVK protrudes through the binding groove

To determine how C-terminal extensions protrude from the F' pocket of MHC I, we bound the extended ligand FVLELEPEWTV**K** (UFP[16-27]) and its binding core FVLELEPEWTV (UFP[16-26]) to HLA-A*02:01, crystallized these two complexes, and solved their structures at a resolution of 1.5 Å and 1.87Å, respectively (*Supplementary file 4*). Both peptides bound in a zig-zag fashion in order to

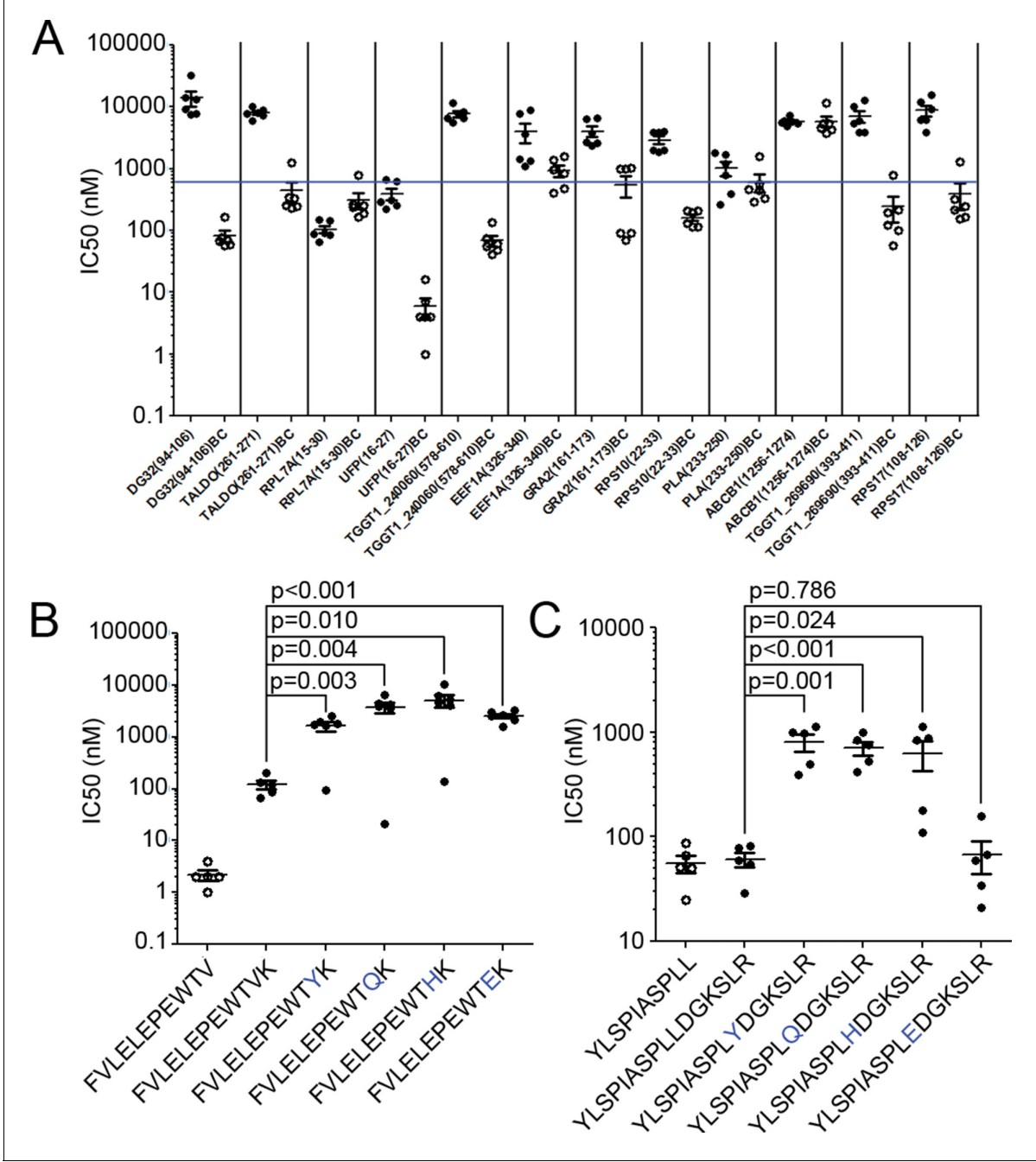

**Figure 7.** Binding affinity of extended ligands and their respective binding cores. (**A**) Measured IC50 of extended peptides (black fill) and the respective predicted binding core (white fill). Blue line denotes 500 nM; binding affinities below this are considered binders. (**B, C**) Mutation analysis of FVLELEPEWTVK and YLSPIASPLLDGKSLR with non-permissive F' pocket residues. Blue letters denote the mutated residue. All data shown are the results of two independent experiments run in triplicate or duplicate. P-values shown are the result of an unpaired two-tailed t-test.

accommodate the core 11 amino acids in the antigen-binding groove (*Figure 9A*). The 11 amino acid core of extended ligand FVLELEPEWTV<u>K</u> interacts with HLA-A*02:01 in an almost identical manner to the shorter peptide, however, both the main chain and side chain of C-terminal Lys12 protrudes out the end of the binding pocket (*Figure 9B*). Electron density for both peptides was very well defined over the entire peptide length (*Figure 9C,D*). The N-terminal and C-terminal amino acid residues of the peptide provide the majority of H-bond and van der Waals contacts

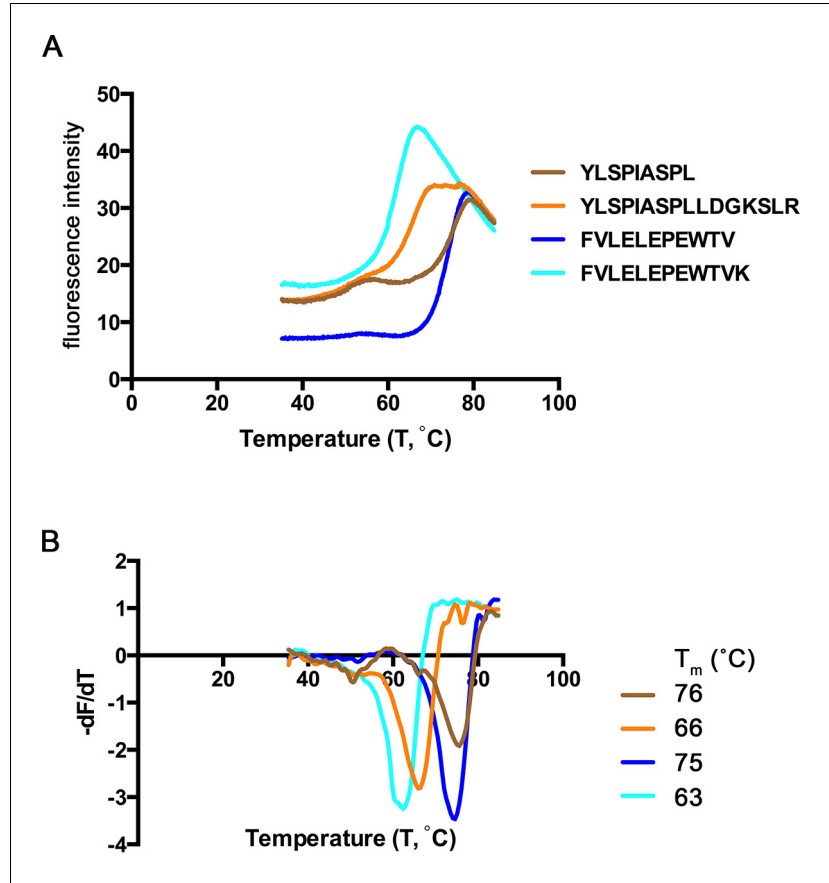

**Figure 8.** Thermal denaturation of extended ligands. (**A**) Raw fluorescence of the melt curve for indicated peptide/HLA-A*02:01 complex. (**B**) First derivative of the melt curve from thermal denaturation experiment for HLA-A*02:01 indicated peptide ligand. The melting temperature for each peptide was calculated from the minima of these curves and is shown in the figure legend.

(*Supplementary file 4*). When attention was shifted to the F- pocket of HLA-A*02:01, with the shorter peptide this pocket was closed by the side chains of Thr80 and Tyr84 with Lys146 of the MHC I providing a lid or cover above the F pocket thereby burying the peptide's C-terminal 11[th] residue underneath (*Figure 9A,E*). However, for HLA-A*02:01 in complex with FVLELEPEWTV<u>K</u>, the side chain of Tyr84 swung up and out by almost 90 degrees (*Figure 9A,F*), opening the binding groove so that a peptide might protrude from the groove at its C-termini. At the same time, Thr80 adopted a different rotamer, further opening the MHC I pocket toward the end of the α1-helix. Lastly, there was a subtle but noticeable increase in the main chain distance between the α 1 and α2-helices at the F' pocket (*Figure 9G*). Together, these structural changes opened the F' pocket and allowed Lys12 of the peptide to protrude from the pocket.

We compared our structures to a previously reported structure of HLA-A*02:01 in complex with a peptide that also extends its C-terminus from the F' pocket (*Collins et al., 1994*). While the authors anticipated the structural changes necessary to allow a peptide to protrude from the F' pocket, the reported structure shows a minor structural change where Tyr84, the key player in opening the F' pocket, rotates slightly but does not swing out, and Thr80 did not change its rotamer. A slight movement of the Lys146 side chain, which appeared to be sufficient to allow the small C-terminal glycine to stick up, was reported (*Figure 9G*). However, in this previously reported model, amino acids other than glycine could not extend from the F' pocket due to steric clashes with the MHC I. Here, we have identified a previously unreported mechanism that allows pathogen-encoded C-terminally extended peptides to protrude out the end of the MHC I binding grove at the F' pocket. The

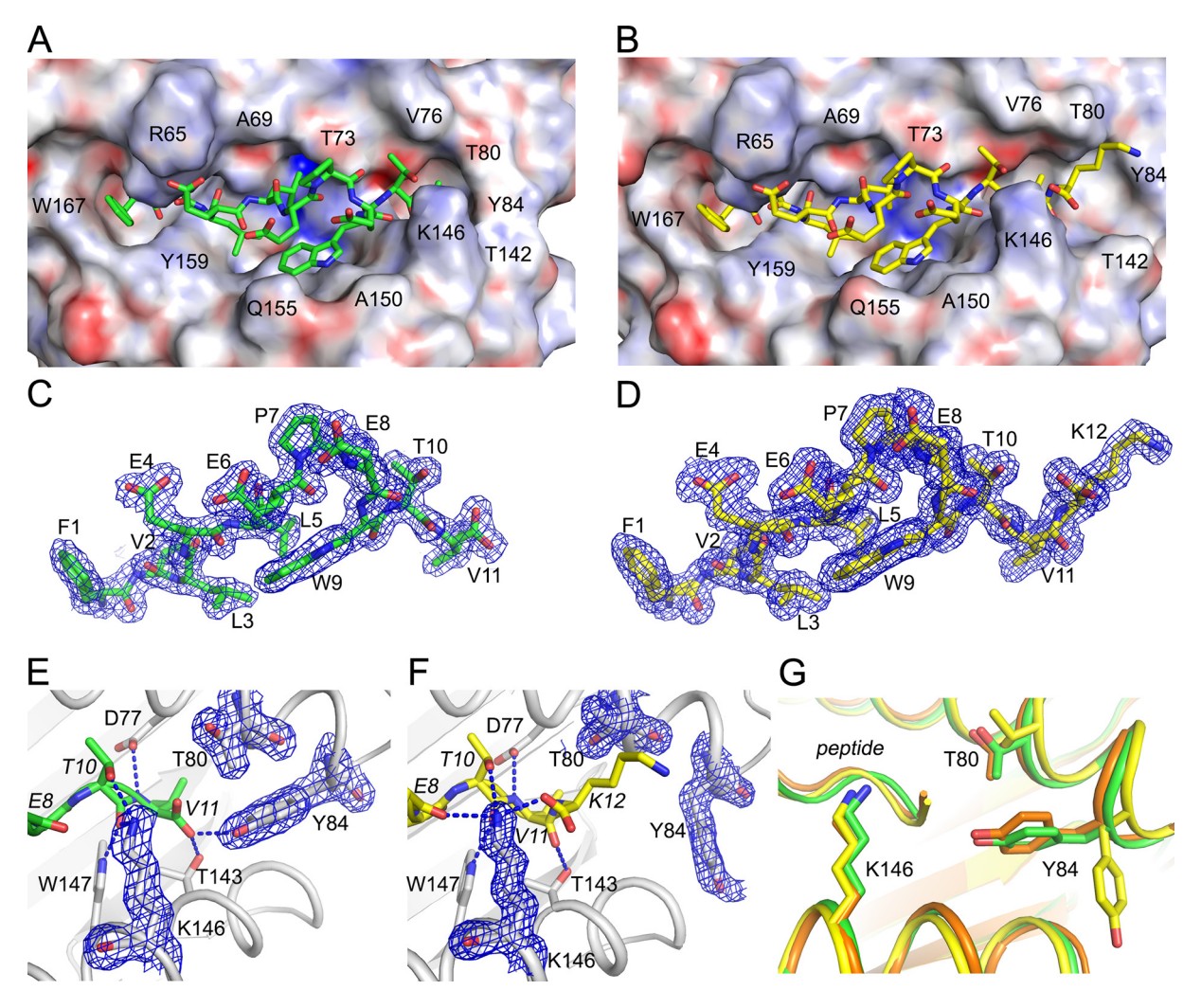

**Figure 9.** Structural details of extended ligand binding to HLA-A*02:01. Binding of core peptide FVLELEPEWTV (A, C, E) and extended ligand FVLELEPEWTVK (B, D, F) to HLA-A*02:01. Peptides are shown as sticks, while HLA-A*02:01 is shown as a molecular surface with electrostatic potential contoured from -30kT/E to +30kT/E (positive charge in blue, negative in red). Peptide FVLELEPEWTV in green, and FVLELEPEWTVK in yellow. 2FoFc electron density is shown as a blue mesh around the peptide (2 Å radius) FVLELEPEWTV (C) and FVLELEPEWTVK (D) and contoured at 1σ Details of peptide binding to the F' pocket of MHC (E, F). MHCI residues that form H-bond interactions (blue dashed lines) with the peptide are labeled. MHC residues that are critical for the F' pocket formation are shown with electron density with same settings as in C and D. (F) Note how Thr80 (T80) and Tyr84 (Y84) change position upon binding of extended ligand FVLELEPEWTVK. Those structural changes are not seen in PDB ID 2CLR (orange) when superimposed with UFP (16–26) and UFP (16–27).

binding cores of these C-terminally extended peptides interacts with MHC I using the same peptide register of canonical MHC I binders.

## Discussion

Proteins secreted by *T. gondii* have been demonstrated to be an important source of immunogenic MHC I peptide ligands (*Grover et al., 2014*). Indeed, the secreted dense granule proteins are the best-characterized source of peptides for presentation by MHC I, and our study confirms these proteins provide a number of MHC I ligands (*Blanchard et al., 2008*; *Cardona et al., 2015*; *Cong et al., 2011*). However, many of the ligands we observe originate from non-secreted proteins, including parasite cytoplasmic proteins. Enrichment in these proteins is unexpected given that most *T. gondii* cytoplasmic proteins are not secreted and their cellular location should sequester them

from the MHC I peptide processing machinery. Further, we identified many hypothetical proteins sampled by MHC I. Proteomic data had previously been reported for all but one of these hypothetical proteins, and our data substantiates that these proteins are expressed by the tachyzoites within the infected host (*Gajria et al., 2008*; *Treeck et al., 2011*; *Xia et al., 2008*). Additionally, a little less than half (10/22) of the hypothetical proteins contain a predicted secretion signal sequence. These data represent a substantial contribution to understanding patterns of *T. gondii* gene/protein expression and demonstrate that secreted as well as non-secreted tachyzoite-expressed proteins are accessible to host antigen presentation processes.

A previous study of MHC I and *T. gondii* indicated that ligands derived from the C-termini of *T. gondii* source proteins are immunodominant (*Feliu et al., 2013*), and it was suggested that a proteolytic insufficiency within infected murine cells meant that only peptides derived from the C-termini of *T. gondii* source proteins would provide ligands of optimal length (*Feliu et al., 2013*). Consistent with this, we found a dramatic C-terminal bias in the *T. gondii* derived ligands. However, it was completely unexpected that host-derived ligands in the infected cell would also display a C-terminal bias. Gene ontology annotation analysis indicated no bias in the cellular location of the source proteins for these C-terminal ligands (data not shown), so it does not appear to be a protein's location within the cell that results in a C-terminal processing bias. How *T. gondii* is mediating this intracellular alteration in antigen processing and presentation is therefore an unsettled question, and it will be interesting to see if protein ligand sampling is also biased in the closely related parasite *Plasmodium falciparum*.

One of the most striking observations in this study is that infection leads to the MHC I presentation of *T. gondii* ligands that are longer than baseline host ligands. Most ligands of considerable length are thought to be anchored to the B' and F' pockets of the MHC I groove via the peptide's P2 and PΩ side chains, respectively. In such models the long peptide ligands exhibit a central bulge in order for the MHC I groove to accommodate the length of the extra residues (*Hassan et al., 2015*; *Tynan et al., 2005*). Consistent with this model, several long *T. gondii* ligands identified follow this P2/PΩ-anchor means of binding to MHC I. Rather unexpectedly, a group of long ligands did not follow this model but instead were predicted to bind via a canonical N-terminal binding core preceding the aforementioned C-terminal extension. A review of published ligand elution data confirms that C-terminally extended peptides are presented by membrane-bound HLA-A*02:01 (*Hassan et al., 2015*; *Chen et al., 1994*; *Scull et al., 2012*), yet the considerable number of extended ligands observed in this in vitro model of *T. gondii* antigen processing and presentation merits future confirmation in vivo. That notwithstanding, structural support for the binding of extended peptide ligands in HLA-A*02:01 can also be found (*Collins et al., 1994*). In this example the nonamer binding core of a calreticulin peptide is extended by a single C-terminal amino acid that fits within the MHC I groove, yet this model was not consistent with *T. gondii* ligands that had as many as 30 C-terminally appended amino acids. In order to resolve this enigma, an HLA-A*02:01 crystal was solved with F<u>V</u>LELEPEWT<u>V</u>K and with a version of this ligand minus the C-terminal K. This led to the unprecedented observation that the C-terminal lysine at P12 displaced the Tyr84 residue at the end of the HLA-A*02:01 binding groove. As such, Tyr84 emerged as a "swinging gate" whereby a longer *T. gondii* peptide extended straight out the end of the HLA molecule at its C-terminus when Tyr84 assumed an alternate up and out position – the open gate. With the shorter peptide the Tyr84 was positioned down and in, assuming the traditional closed-groove orientation. A Tyr84 in the open position is consistent with several extended *T. gondii* ligands and a number of host ligands. This structural configuration is distinct from c-terminal extensions previously reported with C-terminal extension of covalently linked peptides (*Mitaksov et al., 2007*) and predicted configurations (*Hörig et al., 1999*) where the peptide travels up and over Tyr84. Note that Tyr84 modestly rotated to facilitate the reported binding of calretulin to HLA-A*02:01 (*Collins et al., 1994*), and the gate-open displacement of Tyr84 observed here confirms that the MHC I binding pocket is less rigid than previously realized. Importantly, this method of binding appears to be a major mechanism by which *T. gondii* and peptides are bound and presented in contrast to host-derived peptides where this mode of binding significantly less frequent.

The increased frequency of extended peptides following infection suggests that *T. gondii* ligands emerge from a distinct pathway of antigen processing and presentation. Class I MHC peptide ligands are typically derived from the proteasomal degradation of cytosolic proteins, active transport by TAP into the lumen of the ER, further proteolytic trimming in the ER, and chaperone mediated

loading into class I MHC prior to egress to the plasma membrane. In the case of *T. gondii*, parasites are contained within a fusion resistant parasitophorous vacuole (PV), making it necessary that alternative mechanisms contribute to *T. gondii* protein degradation, transport, and trimming prior to presentation. Current evidence shows that *T. gondii* proteins secreted into the PV can be retrotranslocated via the endoplasmic reticulum associated degradation (ERAD) complex from lumen of the PV to the host cytosol where they are routed to MHC I (*Blanchard et al., 2008*; *Feliu et al., 2013*; *Grover et al., 2014*; *Gubbels et al., 2005*). This process of presenting canonical peptide ligands from exogenous non-cytosolic proteins by class I MHC is referred to as cross-presentation (*Blanchard and Shastri, 2010*; *Grotzke and Cresswell, 2015*). The extended *T. gondii* ligands observed here do not seem to fit this model of cross-presentation as their C-terminal extensions suggest a lack of interaction with proteolytic agents of the host cytosol and as many *T. gondii* ligands are derived from proteins not secreted into the PV. Rather, infected cells seem to exhibit a distinct means of cross-presentation, almost as though extended ligands move directly from the PV or the pathogen itself to the host's ER, forgoing exposure to carboxypeptidases found in the cytosol that are otherwise absent from the lumen of the ER. Indeed evidence of a semi-permeable channel between the PV and the ER might explain the presentation of extended peptides (*Goldszmid et al., 2009*). However it is that ligands of unusual length reach class I MHC, future studies of T cell immunogenicity to *T. gondii*, and possibly to other large intracellular pathogens, must factor the distinct environment of the infected cell into the identification of immune epitopes.

In summary, this study reports how antigens encoded by the large intracellular pathogen *T. gondii* are processed and presented by the host cell's MHC I. Studies of peptides that are naturally processed and presented by the MHC I of mammalian cells have historically used healthy, or uninfected, cells to provide the baseline understanding for how antigens are made available for immune recognition. A foundation has emerged whereby peptide ligands of 9 amino acids are enveloped into an MHC I binding groove such that the central portions of the peptide ligand are available for review by adaptive immune receptors, and a legion of structural and functional data support this paradigm. Here we see that thousands of peptide ligands harvested from the MHC I of infected cells also fit this canonical antigen processing and presentation model, but in parallel we observe that alternate and unanticipated mechanisms result from infection and play a role in making *T. gondii* available for immune recognition. That long ligands can extend from their C-termini in a linear fashion out the end of what was previously recognized as a closed MHC I groove was unexpected. Our findings raise a plethora of important questions that must now be addressed, including whether other large intravacuolar pathogens such as *Plasmodium* and *Mycobacterium* species mediate similar changes to MHC I ligand presentation, how infection remodels host cell biology to facilitate the delivery of long extended ligands to MHC I, and the impact that unconventional ligand presentation has on adaptive immune responses to infected cells.

## Materials and methods

### Cell lines and *T. gondii* strains

THP-1 cells acquired from ATCC (ATCC# TIB-202) were cultured in RPMI supplemented with 10% FBS. THP-1 cells were routinely authenticated by HLA typing using sequence based typing at the HLA-A, B, C, and DRB1 loci from an American Society for Histocompatibility and Immunogenetics (ASHI) accredited laboratory (ASHI#03-5-OK-07-1) (*Lanteri et al., 2011*). The reported HLA type of the THP-1 cells are HLA-A*02, -B*15, -C*03, -DRB1*01, -DRB1*15 (*Battle et al., 2013*; *Tsuchiya et al., 1980*). The observed HLA type of the THP-1 cells used in all experiments is HLA-A*02:01, -B*15:11, C*03:03, DRB1*01:01, DRB1*15:01. THP-1 cells were transfected with a soluble form of HLA-A*02:01 as previously described (*McMurtrey et al., 2008*). HLA in the supernatant was measured using a sandwich ELISA using W6/32 as a capture mAb and anti-β2m antibody as a detector. HLA producing cells were subcloned and used for *T. gondii* infection. *Toxoplasma gondii* strain RH expressing GFP was propagated on human foreskin fibroblasts cells acquired from ATCC (ATCC# SCRC-1041) cultured in DMEM supplemented with 10% FBS, glutamine and penicillin/streptomycin. Parasites were released from host cells by passage through a 27-gauge needle (*Wiley et al., 2010*) All host cell lines and parasites were routinely tested for Mycoplasma

contamination with either the MycoAlert Mycoplasma Detection Kit (Lonza, Basel, Switzerland) or Venor GeM Mycoplasma Detection Kit (Sigma-Aldrich, St. Louis MO) and found to be negative..

## HLA production and ligand purification

HLA from uninfected and infected cells were purified as previously described (*McMurtrey et al., 2008*; *Yaciuk et al., 2014*). Briefly, THP-1 cells producing sHLA-A*02:01 were seeded into a hollow fiber bioreactor. For *T. gondii* infection, cells were expanded to confluence and then infected on day 27 with 3.72 x 10$^9$ parasites. Bioreactor supernatant containing HLA was collected and pooled over the course of the seven day infection. HLA was purified from both infected and uninfected cells using antibody affinity chromatography with an anti-VLDL antibody. HLA was eluted in 0.2 M acetic acid and further acidified to 10% acetic acid. Peptide ligands were dissociated from the alpha chain by heating to 75°C for 15 min. Alpha chain and β-2m were separated from the eluted peptides by 3kDa cutoff ultrafiltration.

## Monitoring *T. gondii* bioreactor infection

THP-1 cells were infected with GFP-expressing parasites in the bioreactor and cells were periodically sampled from the extra capillary space of the bioreactor. 1x10$^6$ cells were stained with 1 ug of the pan-HLA class I specific antibody W6/32 labeled with Alexafluor 647 and incubated at room temperature for 30 min to differentiate whole cells from cell debris and parasites. Cells were washed with 1% BSA in PBS three times and then fixed with 1% PFA for 15 min at room temperature. Cells and free parasites were measured using a BD FACS Calibur flow cytometer.

## Two-dimensional LCMS

HLA peptide ligands are were identified with a two-dimensional LCMS system as described (*Yaciuk et al., 2014*). Briefly, peptide pools were fractionated using high pH off-line reverse phase HPLC. Each fraction was dried, resuspended in 10% acetic acid, and placed into an Eksigent NanoLC 400 U-HPLC auto sampler system (Sciex). Approximately twenty percent of each fraction was injected onto a nano-LC column and eluted with a linear acetonitrile water gradient at low pH (*Yaciuk et al., 2014*). Eluate was ionized with a nanospray III ion source and analyzed with a 5600 Triple-TOF mass spectrometer (Sciex). Survey and fragment spectra for all fractions were analyzed using PEAKS (Bioinformatics Solutions Inc) and were searched against NCBInr database using *Homo sapiens* (ID: 9606) or *Toxoplasma gondii* (ID: 5811) taxonomy filters. For *Homo sapiens* searches a 1% FDR was applied and for *Toxoplasma gondii* a 2% FDR was used. All *T. gondii* unmodified peptide sequences were confirmed with fragmentation of a synthetic peptide.

## Source protein analysis

All source proteins derived from *T. gondii* were manually converted from NCBInr format to ToxoDB (www.toxodb.org) format including official protein names and gene symbols. *T. gondii* source protein gene IDs were used as input for the Gene Ontology Enrichment tool (*Gajria et al., 2008*). Enrichments were calculated using *T. gondii* strain GT1, and both annotated as well as predicted terms were considered. Reported p-values are the Bonferronii adjusted p-value.

## HLA-A*02:01 binding predictions

Predicted HLA-A*02:01 binding affinities were generated for all eluted peptides using NetMHCpan-2.8 (*Hoof et al., 2009*). The percentage rank score was used for all analysis. The percentage rank score indicates how strong a peptide's predicted binding affinity is compared to a large pool of naturally occurring peptides. A rank score of 10% indicates that a peptide is amongst the 10% strongest binding random natural peptides for HLA-A*02:01. Peptides with predicted rank scores <=10% were classified as binders, all other peptides were considered non-binders. All non-binders were screened for potential nested HLA-A*02:01 binders by predicting the binding affinity of all overlapping 8–11mers within the eluted peptide sequence. A more conservative rank score <=2% was used to identify nested binders. If multiple nested binders were identified within the same eluted peptide, the nested binder with the strongest predicted binding affinity was selected. Permuted peptides were generated by scrambling the amino acid sequence of the eluted peptide and predictions were preformed in the same manner.

## Binding assay

Assays to quantitatively measure peptide binding to HLA-A*0201 (MHC I) molecules are performed essentially as detailed elsewhere (*Sidney et al., 2001*; *2008*; *Sidney, 2013*). In brief, 0.1–1 nM of radiolabeled peptide is co-incubated at room temperature with 1 μM to 1 nM of purified HLA-A*02:01 in the presence of a cocktail of protease inhibitors and 1 μM β2-microglobulin. Following a two day incubation, HLA-A*02:01 bound radioactivity is determined by capturing the HLA/peptide complexes on W6/32 (anti-class I) antibody coated Lumitrac 600 plates (Greiner Bio-one, Frickenhausen, Germany), and measuring bound cpm using the TopCount (Packard Instrument Co., Meriden, CT) microscintillation counter. Under the conditions utilized, where [label]<[HLA] and IC50 ≥ [HLA], the measured IC50 values are reasonable approximations of the true Kd values (*Cheng and Prusoff, 1973*; *Gulukota et al., 1997*). Each competitor peptide is tested at six concentrations covering a 100000-fold dose range in three or more independent experiments. As a positive control, the unlabeled version of the radiolabeled probe is tested in each experiment.

## HLA-A*02:01 expression and purification for crystal structure

HLA-A*02:01 class I heavy chain ectodomain (residues 1–274) and human β-2 microglobulin (hβ2m, 1–99) were expressed as inclusions bodies and refolded as reported previously (*Garboczi et al., 1994*) with modifications as reported. Briefly, 15 mg of HLA-A heavy chain mixed with 3 mg of peptide (GenScript) was then added to the refolding mix and further stirred at 4°C for 72 hr. Final heavy chain:light chain:peptide ratios were 2.5:1:12 for peptides FVLELEPEWTVK and FVLELEPEWTV. Following refolding, refolding mixture was spun at 50000g to remove any precipitated protein, supernatant concentrated to about 3 ml and loaded onto a Superdex S200 HR16/60 gel filtration column. Fractions containing refolded HLA-A*02:01-peptide complexes were pooled, concentrated to about 10–12 mg/mL and used for crystallization experiments.

## Thermal denaturation assay

HLA-A*02:01-peptide complexes with peptides FVLELEPEWTV, FVLELEPEWTVK, YLSPIASPL and YLSPIASPLLDGKSLR were analyzed for thermal denaturation by differential scanning fluorimetry. HLA-A*02:01-peptide complexes at 100 μM in reaction buffer (20 mM Tris-HCl pH 7.5, 150 mM NaCl) were used as protein stock solution. Each reaction comprised of 1–2 μl protein stock solution, 2 μl of SYPRO Orange dye (100X, Invitrogen) made up to 20 μl in reaction buffer. The experiment was performed in triplicates for individual peptide complexes using a LightCycler 480 (Roche) in a 96-well plate format. A temperature gradient from 20°C - 85°C at steps of 0.06°C/sec and 10 acquisitions/°C was run. The melt curve of the total fluorescence was plotted against the temperature. The first derivative of the melt curve was obtained from raw fluorescence data (temperature differential of absolute fluorescence versus temperature) and plotted as well. The minima in the first derivative of each melt curve, corresponding to the inflection point of the original melt curve, provided the melting temperature ($T_m$) of each protein (*Figure 8B*) (*Walden et al., 2014*).

## Crystallization and data collection

Thin plate-like crystals were obtained for HLA-A*02:01 complex with UFP (16–26) in 30% PEG 5000 MME, 0.1 M Tris-HCl pH 8.0, 0.2 M lithium sulfate and HLA-A*02:01 complex with UFP (16–27) in 30% PEG 4000, 0.1 M Tris-HCl pH 8.0, 0.2 M lithium sulfate. Crystals were obtained by sitting drop vapor diffusion by mixing 0.15 μl protein and 0.15 μl of precipitant at 20°C after 2–4 days. The crystals were flash frozen in cryoprotectant (Reservoir solution: 100% glycerol - 3:1) using liquid nitrogen. Diffraction data for HLA-A2/ UFP (16–26) and HLA-A2/ UFP (16–27) were collected remotely at beamline 7.1 at the Stanford Synchrotron Radiation Light source (SSRL) and processed to 1.8 Å and 1.5 Å resolution, respectively using HKL2000 (*Otwinowski and Minor, 1997*). Phases were obtained using the protein coordinates for HLA-A2 (PDB ID 3MRE) using molecular replacement with Phaser MR (*Storoni et al., 2004*) in ccp4i (*CCP4, 1994*; *Potterton et al., 2003*) and provided unambiguous electron density for both the peptides. Model building was carried out using COOT (*Emsley and Cowtan, 2004*; *Emsley et al., 2010*). Structures were refined using Refmac (*Deckert-Schlüter et al., 1994*) to a final $R_{work}$/$R_{free}$ of 0.164/0.222 for HLA-A2/UFP ( 16–26) (PDB ID 5D9S) and $R_{work}$/$R_{free}$ of 0.195/0.219 for HLA-A2/UFP (16–27) (PDB ID 5DDH).

## Acknowledgements

This work was supported by: A subcontract to WHH and IB from NIH U19 AI062629-05 Coggeshall/
Szeto, NIAID contract number HHSN272201200010C (to BP), DMID-NIAID U01 AI77887, R01 27530
(to RM), and The Research to Prevent Blindness Foundation (to RM).

## Additional information

### Funding

| Funder | Grant reference number | Author |
|--------|------------------------|--------|
| National Institute of Allergy and Infectious Diseases | U01 AI77887 | Rima McLeod |
| National Institute of Allergy and Infectious Diseases | HHSN272201200010C | Alessandro Sette |
| National Institute of Allergy and Infectious Diseases | U19 AI062629-05 | Ira J Blader William Hildebrand |

The funders had no role in study design, data collection and interpretation, or the decision to
submit the work for publication.

### Author contributions

CM, MN, IJB, Conception and design, Acquisition of data, Analysis and interpretation of data, Draft-
ing or revising the article; TT, RM, DMZ, WH, Conception and design, Analysis and interpretation of
data, Drafting or revising the article; TS, SGR, Acquisition of data, Analysis and interpretation of
data, Drafting or revising the article; TK, WB, KJ, Acquisition of data, Analysis and interpretation of
data; AS, Conception and design, Drafting or revising the article; BP, Conception and design, Analy-
sis and interpretation of data

### Author ORCIDs

Curtis McMurtrey, http://orcid.org/0000-0003-4540-4140
Thomas Trolle, http://orcid.org/0000-0003-0762-2198

## Additional files

### Supplementary files

• Supplementary file 1. Species ambiguous ligands.

• Supplementary file 2. Identified unambiguous *T. gondii* ligands.

• Supplementary file 3. Predicted extended ligands and binding cores. Bold sequences indicate the
peptides selected for the binding assay.

• Supplementary file 4. Data collection and refinement statistics for crystal structures.

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
