## [Decision Letter]

Thank you for submitting your work entitled "*Toxoplasma gondii* Peptide Ligands Open the Gate of the HLA Class I Binding Groove" for consideration by *eLife*. Your article has been favorably evaluated by Tadatsugu Taniguchi (Senior editor) and three reviewers, one of whom, Michael S. Gilmore, is a member of our Board of Reviewing Editors.

The reviewers have discussed the reviews with one another and the Reviewing Editor has drafted this decision to help you prepare a revised submission.

Summary:

General assessment:

The reviewers agreed that this manuscript describes has the potential to significantly advance our understanding of the repertoire of *T. gondii* peptides displayed by MHC Class I HLA-A2, and the extend our understanding of the nature of peptides bound and the manner in which they bind.

Central conclusions:

1) A comprehensive catalogue of *T. gondii* peptides presented by HALA-A2 is provided, expanding the current understanding by orders of magnitude.

2) Dense granules were found not to be the main source of *T. gondii* peptides presented, in contrast to current dogma. Many peptides derived from non-secreted cytoplasmic proteins.

3) Curiously, about a third of peptides presented were identical to counterparts in human proteins.

4) Unexpectedly, *T. gondii* peptides overall were about 40% longer than predicted – some much longer.

5) Long peptide binding appears to involve novel interactions with the HLA molecule. Crystal structural analysis showed that rather than forming a bulge, the large *T. gondii* peptides induce a conformational change in HLA that allows the C-terminal extension to protrude from the cleft.

Essential revisions:

The following points were raised during review that must be adequately addressed before the paper can be accepted.

1) The conformational change involving Tyr84 in the FVLELEPEWTVK complex raises questions about the relative stability of the complex and/or the relative dissociation rates of the peptides, and the extent to which they may be functionally important. This could be resolved for example by measuring the stability of the complexes (e.g. melting temperature) or the dissociation rates of the peptides.

2) All of the peptide ligands are eluted from an engineered, artificial HLA-A2 molecule that is designed to be secreted from the THP cells. How does this change the dynamic between the sHLA and the peptide loading and editing machinery? How have the authors ruled out that the longer *T. gondii* HLA-peptide ligands are not an artifact associated with the secreted HLA protein?

3) Why does the distribution of *T. gondii* derived peptides skew toward longer peptides than the host derived peptides? (One reviewer noted that the difference in the distributions of host-derived ligands between infected and uninfected cells seem to be derived from a single experiment – is there more data to support this?). The potential for cross-presentation is not discussed, and this issue should be addressed.

---

## [Author Response]

Essential revisions: The following points were raised during review that must be adequately addressed before the paper can be accepted. 1) The conformational change involving Tyr84 in the FVLELEPEWTVK complex raises questions about the relative stability of the complex and/or the relative dissociation rates of the peptides, and the extent to which they may be functionally important. This could be resolved for example by measuring the stability of the complexes (e.g. melting temperature) or the dissociation rates of the peptides.

This is an outstanding point – characterizing the stability of these extended peptide/HLA complexes is positioned to further support the conclusions reached here. To address this point we have now performed thermostability assays on the two extended peptides YLSPIASPLLDGKSLR and FVLELEPEWTVK. We also completed a thermostability analysis with their respective binding cores. We find that the two extended ligand complexes are quite stable, having melting temperatures consistent with other reported HLA-A2 ligand complexes. Moreover, the core peptides, although they were never found to be presented, were considerably more stable, with a Tm of at least 10˚C above their extended counterpart. Thus, the extended ligands are stable in solution as compared to canonical as well as bulging peptide ligands while the core alone substantially increases the stability of the complex. In retrospect, it is likely that a highly stable core is key to the stability of the extended ligand. These data have been added as a new paragraph in the Results section of the manuscript along with Figure 8 that depicts these data.

2) All of the peptide ligands are eluted from an engineered, artificial HLA-A2 molecule that is designed to be secreted from the THP cells. How does this change the dynamic between the sHLA and the peptide loading and editing machinery? How have the authors ruled out that the longer T. gondii HLA-peptide ligands are not an artifact associated with the secreted HLA protein?

The reviewers raise a question that we now address in the revised manuscript. While it is possible that the extended peptides are an artifact associated with the secreted HLA protein, there are previous studies demonstrating that secreted HLA proteins are not significantly altered in their trafficking or in their peptide repertoires (Skull et al 2012). Specifically, a study comparing secreted HLA-A*02:01 and membrane-bound HLA-A*02:01 shows that the secreted HLA traffics and matures in the same manner as membrane-bound HLA, albeit at different rates. Furthermore, there is little to no difference in the ligand repertoires of secreted and membrane-bound HLA (Skull et al 2012). In regards to the presentation of longer ligands, these data have been quietly emerging for some time, yet nobody has characterized them to the extent we have here. One analysis of ligands eluted from membrane-bound HLA ligands shows the presence of extended ligands, although the frequency of extended ligands was not addressed (Skull et al 2012, Chen et al 1994, and Hassan et al 2015). Reports of ligand extension in mouse MHC class I provide further evidence of extended ligands (Horig et al 1999). Simply put, membrane preparations of HLA provide relatively small quantities of eluted ligand for characterization. Furthermore, extended peptides remain hidden in membrane preparations of HLA because ligands from 3-6 HLA are usually characterized simultaneously such that it is difficult to assign a binding core and HLA restriction to the extended ligands and are often left out of the analysis. Secretion provides a single HLA so that extended ligands can clearly be assigned to their restricting class I molecule. Finally, the ligands discovered in secreted HLA have successfully been developed into viral vaccines and human melanoma vaccines (1, 2), demonstrating the biological relevance of the secreted HLA model. In summary, there is a considerable body of data supporting the use of secreted HLA to more fully appreciate ligand repertoires. Secreted HLA reflects membrane-bound HLA as demonstrated in both biochemical and clinical studies, and secreted HLA provides perhaps the only approach to clearly assign extended ligands to their corresponding HLA. We nonetheless included, in the revised manuscript, a Discussion statement suggesting that future studies using very large scale in vivomembrane-bound HLA will more definitively demonstrate these data.

*3) Why does the distribution of T. gondii derived peptides skew toward longer peptides than the host derived peptides? (One reviewer noted that the difference in the distributions of host-derived ligands between infected and uninfected cells seem to be derived from a single experiment* – *is there more data to support this?). The potential for cross-presentation is not discussed, and this issue should be addressed.*

Again, we are in complete agreement with the referees. To address this concern in the revised manuscript we have added a Discussion paragraph that briefly reviews cross-presentation and that expounds on possible cross-presentation mechanisms that might explain a possible mechanism by which extended *T. gondii* ligands access the class I HLA peptide loading complex. We feel that this added paragraph succinctly addresses the potential role of cross-presentation in *T. gondii* infection.

References:

1) Carreno BM, et al. (2015) Cancer immunotherapy. A dendritic cell vaccine increases the breadth and diversity of melanoma neoantigen-specific T cells. Science 348(6236):803-808.

2) Kim S, et al. (2010) Single-chain HLA-A2 MHC trimers that incorporate an immundominant peptide elicit protective T cell immunity against lethal West Nile virus infection. J Immunol 184(8):4423-4430.